# A Health Guidance App to Improve Motivation, Adherence to Lifestyle Changes and Indicators of Metabolic Disturbances among Japanese Civil Servants

**DOI:** 10.3390/ijerph17218147

**Published:** 2020-11-04

**Authors:** Naoko Takeyama, Michiko Moriyama, Kana Kazawa, Malinda Steenkamp, Md Moshiur Rahman

**Affiliations:** 1Division of Nursing Science, Graduate School of Biomedical and Health Sciences, Hiroshima University, Hiroshima 734-8553, Japan; morimich@hiroshima-u.ac.jp (M.M.); kkazawa@hiroshima-u.ac.jp (K.K.); Malinda.steenkamp@protonmail.com (M.S.); 2Torrens Resilience Institute, Flinders University, Adelaide, SA 5001, Australia

**Keywords:** metabolic disturbances, lifestyle changes, ICT application, health guidance, motivation, adherence, workers’ health

## Abstract

We investigated whether an Information and Communication Technology (ICT) application (app) motivated to increase adherence to lifestyle changes, and to improve indicators of metabolic disturbances among Japanese civil servants. A non-randomized, open-label, parallel-group study was conducted with 102 participants aged 20–65 years undergoing a health check during 2016–2017, having overweight and/or elevated glucose concentration. Among them, 63 participants chose Specific Health Guidance (SHG) and ongoing support incorporating the use of an app (ICT group) and 39 individuals chose only SHG (control group). Fifty from the ICT group and 38 from the control group completed the study. After completing the 6-month program, the control group showed a significant decrease in body mass index (*p* = 0.008), male waist circumference (*p* < 0.001), systolic blood pressure (BP) (*p* = 0.005), diastolic BP (*p* < 0.001), and glycated hemoglobin (HbA1c) (*p* < 0.001), and increase in high-density lipoprotein (HDL) cholesterol (*p* = 0.008). However, the ICT group showed a significant decrease in male waist circumference (*p* < 0.001), diastolic BP (*p* = 0.003), and HbA1c (*p* < 0.001), and increase in HDL cholesterol (*p* = 0.032). The magnitude of change for most indicators tended to be highest for ICT participants (used the app ≥5 times/month). Both groups reported raised awareness on BP and weight. The app use program did not have a major impact after the observation period. Proper action requires frequent use of the app to enhance best results.

## 1. Introduction

Lifestyle-related diseases, including metabolic syndrome, are increasing worldwide. Persons with metabolic syndrome have a higher risk of diabetes, atherosclerotic cardiovascular disease, non-alcoholic fatty liver disease, and potentially linked to morbidity and mortality [1,2]. Lifestyle improvement is the key to managing the condition. All five indicators for metabolic syndrome (i.e., larger waist circumference, elevated blood pressure (BP), raised triglyceride levels, reduced high-density lipoprotein (HDL) cholesterol, and raised fasting glycaemia) [3] improve with therapeutic changes to diet and exercise [4].

Many studies have focused on how to motivate and maintain therapeutic behavioral changes in persons with metabolic syndrome as sustained motivation and adherence to changes are crucial [4]. A team-based, interactive approach that keeps individuals accountable and provides regular personalized feedback have been found to foster persistent and effective behavioral change [4]. Evidence indicates that tailored interventions are important [5]. Tailored interventions are those where the needs and abilities of an individual are assessed through an initial evaluation and advice is then personalized to that person. Personal motivation is enhanced if the customized feedback is perceived as appropriate and applicable, and interventions are more effective when individuals have a sense of ownership in the therapeutic process [6,7]. Studies on tailored interventions are increasing and reflect parallel advances in the use of information and communication technology (ICT) in interventions [8].

Collecting patient data relevant to interventions have expanded from face-to-face interactions (such as motivational interviews or counselling) and telephone conversations to computer programs, smartphone applications, websites, emails, and short message services (SMS) [9,10]. Smartphone applications and websites allow for the automated collection of patient data which can be used to customize interventions and advice [11]. Using ICT also allows for the use of several intervention elements in one program, such as education, feedback, and self-monitoring [12], often in a way that is user-friendly and acceptable for the intended audience [9]. Several studies have found that tailored interventions using ICT are effective in addressing metabolic syndrome [12,13].

In Japan, all health insurers, whether a national or a social insurance, are required to screen the insured and their dependents aged 40–74 years for risk factors of metabolic syndrome from 2008 [14]. Insurers must also provide lifestyle improvement counseling (referred to as Specific Health Guidance—SHG) for non-medicated persons who have elevated risk factors of metabolic syndrome [15,16]. Diagnostic criteria of metabolic syndrome in Japan were defined by the Diagnostic Criteria Review Committee of Japan (Appendix A) [17]. The selection criteria for SHG were adopted by the following measures. There are two types of SHG in this national program [14]. The first is the Intensive Health Guidance (IHG) for patients with ≥2 markers with abdominal obesity or those with ≥3 risk factors without abdominal obesity, but with a body mass index (BMI) >25 kg/m^2^. The second type is the Motivational Health Guidance (MHG) offered to those with 1 risk factor with abdominal obesity or 1–2 markers without abdominal obesity but with a BMI >25 kg/m^2^. Both IHG and MHG comprise an initial motivational counseling session with a final evaluation after 6 months. At the initial session, participants are provided with information about their condition, and their lifestyles are reviewed using their screening results. They are advised to set achievable personalized behavioral goals, that include increased exercise (e.g., walking an extra 10 min), self-monitoring (wearing a pedometer), and adjusting their diet. The initial session is delivered by a trained health-care professional (physician, nurse, or dietitian) either as a one-on-one interview or in a group format. Participants in the IHG program are followed up for 3–6 months through e-mails, phone calls, and/or in-person/group sessions, while MHG participants do not receive continuous support. The programs are considered completed when participants received a specific amount of consultation time (IHG) and when the 6-month evaluation is done (both IHG and MHG) [14]. This scheme provides a framework to support the improvement of workers’ health and the prevention of future disease development, which in turn contributes to increased productivity.

In this study, we investigated how the SHG scheme can be further improved to aid workers’ health. The aim was to investigate whether an ICT health guidance application (app) motivated to increase adherence to lifestyle changes among Japanese civil servants with risk of metabolic disturbances. We also examined the changes in indicators of metabolic disturbances related to lifestyle changes made by the participants.

## 2. Materials and Methods

The study was conducted from September 2016 to March 2017 and was a collaborative project between Hiroshima Prefecture and Hiroshima University. The development and implementation of the ICT app was realized with the cooperation of Hiroshima Prefecture and its medical association. The researchers operated within the existing framework of SHG employed by Japanese health insurers.

### 2.1. Study Participants

The study population was civil servants insured by the Mutual Aid Association of Hiroshima Prefecture. Potential participants were those who underwent an annual health check during the study period. Since Hiroshima Prefecture purposefully expanded the target population with not only metabolic syndrome but also at risk of metabolic disturbances for disease prevention, the criteria for this project were developed. Eligibility criteria were clients aged 20–65 years; for age 40 years and older, those who were overweight or obese (defined as waist circumference: ≥85 cm for males and ≥90 cm for females and/or with a BMI ≥ 25 kg/m^2^) and/or who had an elevated glucose concentration (fasting blood glucose (FBG) ≥ 100 mg/dL or glycated hemoglobin (HbA1c) ≥ 5.6%); participants younger than 40 years were included in the study, (i) if their BMI ≥ 25 kg/m^2^, and/or (ii) if their FBG ≥ 100 mg/dL or HbA1c ≥ 5.6%, (iii) if their urinary glucose ≥ trace (±, approximately 50 mg/dl) (Appendix A). This is because being overweight/obese at a younger age increases the risk of developing metabolic syndrome.

Participants were excluded if they were undergoing treatment for metabolic syndrome or other diseases, such as diabetes, cardiovascular diseases, or cancer. Potential participants at risk of metabolic syndrome were approached by the health insurer at the time of their annual health check and enrolled in the study if they were interested, met the eligibility criteria, and gave informed consent.

Of the 6518 people in the study population, 355 (5.4%) expressed an interest to participate, and allowed the researchers to obtain their health check results. After the researcher matched their results to the eligibility criteria, only 102 (28.7% of 355) met the criteria, as shown in Figure 1. Of those, 63 (61.8% of 102) elected to use the ICT app (ICT group). The ICT group received the equivalent of the IHG program (i.e., personal follow-up by email, phone, etc.), irrespective of whether they fell into the IHG or MHG group, as shown in Figure 2. The 39 participants who chose not to use the ICT app (the control group), received either usual IHG (initial session plus midterm interview) or the MHG program (only initial session), as shown in Figure 2. In the ICT group, 1 person withdrew consent and 8 did not use the app. Data was missing for another 4 participants resulting in total of 50 participants in the ICT group, as shown in Figure 1. In the control group, 1 person was excluded because of missing data.

### 2.2. Study Protocol

After agreement to take part in the study, the participants received a one-on-one interview with a health educator from the insurer who provided information about metabolic syndrome, advice on the impact of making lifestyle changes to improve the condition, and encouragement on how to make these changes. The latter included specific guidance on diet changes, increased exercise, and the importance of adherence to changes. The educator also explained the role of continued self-monitoring of BP, heart rate (HR), body weight (BW), and number of steps per day and explained the role of the ICT app to assist them. During the 6 months follow-up, participants received communication from health instructors by either e-mail or telephone, as shown in Figure 2. This included conversations around progress and achievement of goals, as well as additional advice where needed regarding diet, exercise, etc. The progress of the ICT group was also reviewed at around 3 months, as shown in Figure 2. The metabolic indicators were collected by health professionals at baseline and at 6 months (the end of SHG). Before and after SHG, participants visited the same designated licensed health check laboratories, and health professionals did data collection of fasting blood sample (8 mL) and vital signs. BW was measured with light clothes, and BMI was calculated as weight in kg divided by height in m^2^. Waist circumference was measured by placing a tape horizontally around the abdomen at the level of the umbilicus while standing. BP was measured twice at sitting position by the digital BP machine approximately after 5 min to calm down, and lower data was used.

The ICT app called “Hiroshima Health Note” was developed in the electronic network system called the “Hiroshima Medical Network” made by Hiroshima Prefecture and its Medical Association. The app has built-in features to monitor HR, BW, and daily steps, and a feature to record BP was added. Participants could access the application either as a smartphone application or through a website. Participants’ HR, BW, and BP were recorded at the first interview and they were encouraged to continue self-monitoring during the following 6 months. For most participants, HR and BP were measured in the workplace, but some participants had the equipment to do so at home. Their weight was either measured at home or in the workplace. Participants were provided with a pedometer to measure their daily steps. They received auto-generated messages from the ICT app weekly that reminded them to measure and record their BP, HR, BW, and daily steps. The reminders also encouraged participants to review their own data and to continue with behavioral changes. When participants stopped self-monitoring, an auto-generated message to encourage self-measurement was sent. Participants’ data were transmitted to the health instructors every time they measured as on an ongoing basis, as shown in Figure 3, and participants received an individualized weekly summary. Participants in the ICT group received a call from a health educator if their BP, BW, and/or HR were higher than their baseline or their measurements exceeded clinical guidelines.

At the end of the 6 months, a questionnaire was sent to both the ICT and control groups. The ICT participants were asked about their frequency of using the app, reasons for use, and overall satisfaction with it. Both groups were asked about their awareness of metabolic risk factors and their satisfaction with the SHG program.

### 2.3. Outcome Measures

Outcome measures were: indicators of metabolic disturbances (BP, waist circumference, and BMI, triglycerides, HDL, low-density lipoprotein (LDL), HbA1c); behavioral data (frequency of ICT app use: low frequency (<5 times/month) and high frequency (≥5 times/month)); and activity data (number of daily steps).

### 2.4. Statistical Analysis

The final analysis included 50 participants from the ICT group and 38 from the control group, as shown in Figure 1. Data were analyzed using the independent *t*-test or Chi-square test. Comparisons of changes in values for both groups at baseline and at 6 months were performed using the paired *t*-test or Wilcoxon signed-rank test. The Kruskal–Wallis test was performed after the normality test for the test according to the frequency of use of ICT applications. The results of the questionnaire conducted after the intervention were analyzed using the Mann–Whitney U test.

Participants were categorized as having “maintained” results, if they had normal values at baseline and normal values at 6 months. Where they had abnormal results for any indicator at baseline and normal results at 6 months, the intervention was judged as “effective”. Where participants had normal values at baseline, but values outside the normal range at 6 months follow-up, the intervention was judged as “ineffective”. Statistical significance was set as two-sided p-values of less than 0.05. All statistical analysis was done with IBM SPSS Statistics for Windows, v22.0 (IBM Corp., Armonk, NY, USA).

### 2.5. Ethical Considerations

The Epidemiological Ethical Committee of Hiroshima University approved this study (E-701-3). Participants were given an opportunity to opt out after receiving health guidance. Informed consent was obtained, and confidentiality was maintained, and results presented to protect anonymity.

## 3. Results

Table 1 shows that the average age was 50.6 years (standard deviation (SD) ± 6.0 years) for the ICT group and 49.5 years (SD ± 8.2 years) for the control group (*p* = 0.901). There were more women in the control group (21.1%) than in the ICT group (10.3%; *p* = 0.146). At baseline, there were no statistically significant differences between ICT and control groups for any of the indicators of metabolic disturbances, as shown in Table 1.

After the completion of the 6 months program, the control group showed small, but statistically significant improvement in six indicators compared to baseline values, i.e., BMI (*p* = 0.008), male waist circumference (*p* < 0.001), systolic BP (*p* = 0.005), diastolic BP (*p* < 0.001), HDL (*p* = 0.008), and HbA1c (*p* < 0.001), as shown in Table 1. In the ICT group, similar small changes were observed regarding male waist circumference (*p* < 0.001), diastolic BP (*p* = 0.003), HDL (*p* = 0.032), and HbA1c (*p* < 0.001). There were no significant differences when the magnitude of change in measurements for the control group were compared to those of the ICT group, as shown in Table 1.

Of the ICT group, 47 out of 50 participants used the app to monitor their weight and number of daily steps, while 38 used it to record their BP, as shown in Table 2. The number of steps were the most often monitored measurement, with the 47 participants using it on average 15 times/month. Of the 47 that monitored weight, about 75% used it for an average of 9 times/month. Of the 38 that monitored BP, about 80% used it about 7 times/month, as shown in Table 2.

There were some significant changes in indicators of metabolic disturbances when the control group, as well as the low and high frequency ICT users, were compared, as shown in Table 3. For the 38 who monitored their BP, there were significant changes in BMI (*p* = 0.007), waist circumference (*p* < 0.001), triglyceride levels (*p* = 0.007), and HbA1c (*p* = 0.001). For the 47 participants who monitored weight, there were significant changes in waist circumference (*p* = 0.010) and HbA1c (*p* = 0.010). There were no significant differences in any of the indicators for those who monitored their daily steps, as shown in Table 3. The table also shows that the amount of change for most indicators, even for those where statistical significance was not reached, tended to be highest for the ICT participants who used the app more than 5 times/month, followed by the control group. The amount of change was generally lowest for the ICT participants who used the app <5 times/month, as shown in Table 3.

To examine the effectiveness of the program, the participants in each group were classified into the “maintained” group, “effective” group, and “ineffective” group, and the combination of “maintained” and “effective” groups were compared with the “ineffective” group. There were no statistical significances observed in any indicators. However, the ICT group showed nearly 40% better results on systolic BP, and around 10% improvement in diastolic BP, LDL, and HbA1c compared to the control group, as shown in Table 4.

Of the ICT group, 41 (82.0%) completed the questionnaire and of these, 12% were women. A total of 27 participants (71.1%) in the control group responded to the questionnaire and 18.5% were female. At the 6-month mark, more than two-thirds of both the ICT and control groups indicated that they followed a diet suitable for preventing lifestyle-related diseases, as shown in Table 5.

For the ICT group, 78.1% reported that they regularly exercised, compared to 59.2% of the control group. High proportions of both the ICT group (87.8%) and the control group (92.6%) were aware of their average BP value, as well as of their average body weight, i.e., 95.1% for the ICT group and 92.5% of the control group. However, awareness was much lower for blood glucose with 41.5% of the ICT group and 40.7% of the control group. Roughly half of both groups rated the program they were in as effective for continuous lifestyle improvement, but the proportion was slightly higher for the ICT group (53.6%) compared to 48.1% of the control group.

## 4. Discussion

This study explored whether a health guidance app resulted in sustained motivation, adherence to lifestyle changes, and subsequent changes in indicators of metabolic disturbances among civil servants with risk of metabolic syndrome in Hiroshima Prefecture. The ICT group (who had increased contact with health educators and used the app during the 6-month program) showed significant improvement in four measurements: male waist circumference, DBP, HDL, and HbA1c. However, there were also improvements in six measurements for the control group. These measurements were: BMI, male waist circumference, both systolic and diastolic BP, and HbA1c. The magnitude of change between the two groups did not differ significantly. Of note is that the magnitude of changes for both groups were small, e.g., male waist circumference changed by about 3 cm for both groups. Similarly, the HbA1c measurement decreased by about 0.2% for both groups, but this change resulted in the average for the ICT group moving into the normal range. Although the changes are small, a reduction in waist circumference of 3 cm has been shown to be effective in preventing metabolic syndrome. Some studies indicate that there might be a benefit in achieving even small changes. For example, one study of 25 men aged 24–65 years with metabolic syndrome found a small reduction in waist circumference together with weight loss appeared to magnify the impact of the Mediterranean diet on markers of inflammation [18]. Another study using 9-year follow-up data on insulin resistance involved a cohort of 1868 men and 1939 women aged 30–64 years [19]. The authors found that reducing waist measurements by only 3 cm (taking changes to BMI into consideration) had a significant beneficial effect on metabolic syndrome in women.

Hill (2009) suggests a paradigm shift to address the obesity epidemic. He recommends that efforts should, in the first instance, focus on promoting small lifestyle changes and on reducing gradual excessive weight gain, rather than focusing primarily on weight loss or preventing obesity [20]. The latter usually involves making dramatic changes at an individual level which are difficult to sustain and that are often countered by societal pressures of excessive energy consumption (e.g., large portion sizes and free refills of sugary drinks in American restaurants) [21]. Small changes are more feasible to achieve, can be maintained in the longer term, and minor changes in diet and/or in physical activity might be adequate to stop gradual weight gain over time [20]. Success in achieving such changes could lead to increased self-efficacy at the individual level and could encourage people to make additional small changes [20].

Regarding the use of the app, the number of daily steps was the most frequently recorded indicator (about 15 times/month) and just less than 80% of participants measured their steps 5 times or more per month. Less than 50% of participants measured their weight on a regular basis, while about 60% recorded their BP more than 5 times/month. This might reflect how convenient it was to record measurements. Daily steps were measured with a pedometer, which did not require much effort on the part of the participants. However, BP measurements usually necessitated participants to go to specific stations or locations at their workplace where BP could be measured. Similarly, changes in weight usually take about a week to show on the scale. The BMI changed by less than 0.40 for both groups and this slow rate of change might have lessened participants’ motivation to keep weighing themselves.

For most of the indicators, the amount of change was highest for the ICT participants who used the app more than 5 times/month, followed closely by the control group. The magnitude of change was consistently smallest for the low frequency use ICT group. This was applied to most measurements, irrespective of which indicator was recorded or whether statistical significance was reached. This seems to suggest that participants found different types of interventions helpful. A previous study found that insured persons receiving SHG had significant improvements in their metabolic syndrome profiles with smaller proportions needing pharmacological treatment over a 3-year period when compared to people who did not receive SHG [14]. Our study seemed to confirm that SHG had a positive impact on the control group. The more intensive ICT intervention also delivered improvements, but these were not dramatically different from the control group. In this study, participants chose the type of intervention they preferred. Our study indicates that “not one size fits all” and that a higher intensity program does not necessarily translate into better results. Clearly, participants’ preferences play an important factor in the effectiveness of the intervention. A more intensive program appears to hinder adherence if the results of the low frequency use group are considered. This group might have been of the opinion that they would do better with a more intensive program and the use of the app. However, it might have been that they did not like the program and/or they did not like to use the app. This might have led to them not adhering to the program as much.

A recent systematic review on effective lifestyle modification strategies for metabolic syndrome included 28 randomized control trials with a total of 6372 patients [4]. Of these, eight trials found improvement in risk factors of metabolic syndrome after a 1-year follow-up. The authors indicated that team-based, interactive approaches with high-frequency contact with motivated participants made the largest and most lasting impact. The review also looked at the role of technology and indicated that these were useful tools in achieving lifestyle change, but ineffective when compared with personal contact [4]. Another review of 35 studies assessed different intervention delivery methods [5]. The authors indicated that, to maximize the effect of tailored interventions in lifestyle management, the stage of change of the individual must be identified and intervention elements should be based on the readiness of the individual to make changes. The use of ICT, especially the use of mobile technology, does provide the opportunity to collect patient data and provide tailored interventions in a timely fashion. However, insights into the use of ICT in health care in Japan by gender and different generations need further exploration. This study pointed to the issue that women might be less inclined to make use of ICT in managing their health.

## 5. Limitations

There were a number of limitations in this study. The sample size was relatively small, participants were not randomized, and the follow-up period was short (6 months). Participants were also not diagnosed as “patients”, which could have led to less commitment to change on the part of the participants to make and continue changes. The study participants were selected from civil servants of Hiroshima Prefecture. The results might therefore not be generalizable.

## 6. Conclusions

This study found that a once-off motivational interview and a more intensive program of health guidance to address metabolic disturbances in Japanese civil servants show promising results and that the use of ICT can be of benefit to some individuals. However, there are differences in preferences for types of interventions and ICT elements might be a deterrent to some. The interventions need to be framed within a broader context of encouraging small, but consistent changes within society.

## Figures and Tables

**Figure 1 ijerph-17-08147-f001:**
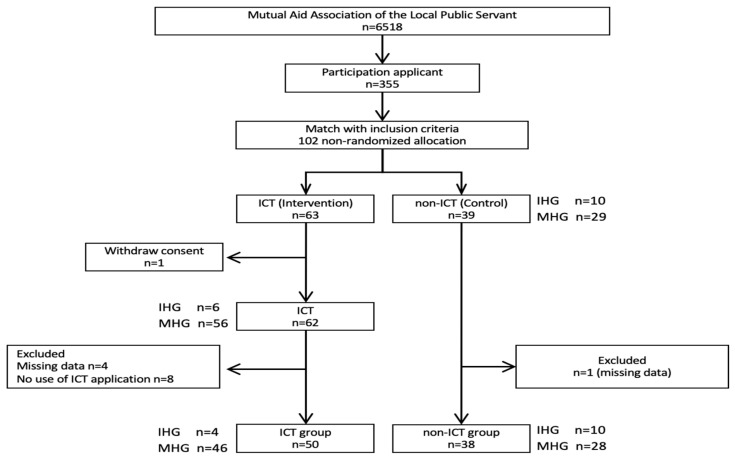
Enrolment of study participants. Note: ICT: information and communication technology; IHG: Intensive Health Guidance; MHG: Motivational Health Guidance.

**Figure 2 ijerph-17-08147-f002:**
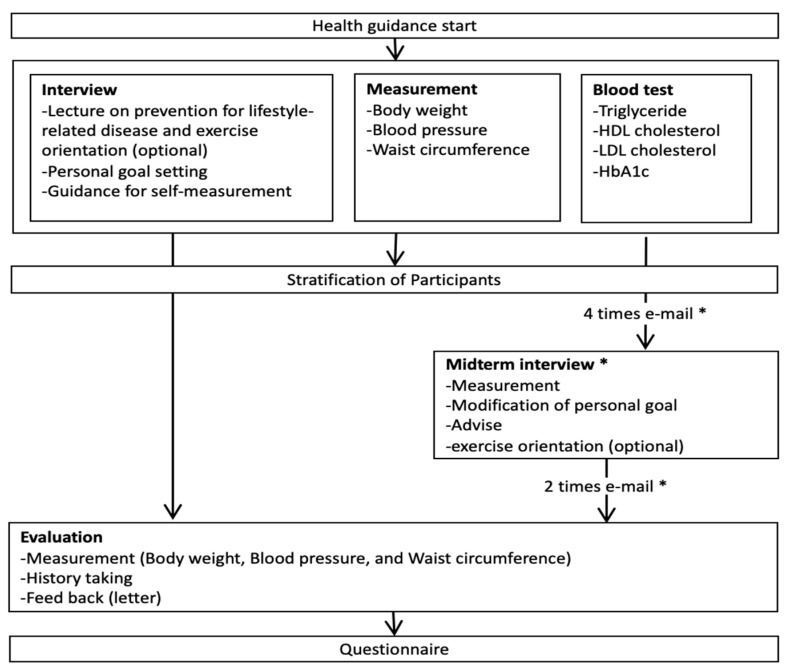
Health guidance program in this study. Note: * IHG program only; HDL: high-density lipoprotein; LDL: low-density lipoprotein; HbA1c: glycated hemoglobin.

**Figure 3 ijerph-17-08147-f003:**
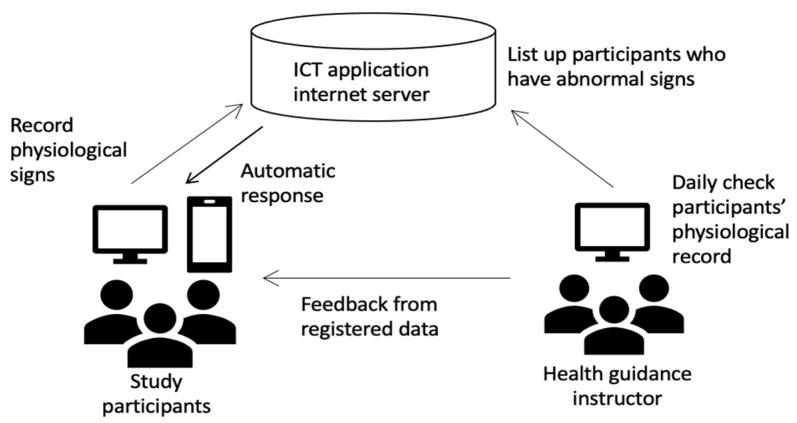
Health guidance process. Note: ICT: information and communication technology.

**Table 1 ijerph-17-08147-t001:** Comparing the ICT and control groups at baseline and at 6 months.

Indicators of Metabolic Disturbances	Baseline Assessment			Assessment at Completion of Intervention 6 Months				Change Between Control and ICT Groups		
	Control Group (N = 38)	ICT Group (N = 50)	*p-*Value ^1^	Control Group (N = 38)	*p-*Value ^2^	ICT Group (N = 50)	*p-*Value ^3^	Control Group (N = 38)	ICT Group (N = 50)	*p-*Value ^4^
	Mean ± SD	Mean ± SD		Mean ± SD		Mean ± SD		Mean ± SD	Mean ± SD	
Body mass index (BMI), kg/m^2^	25.3 ± 3.9	25.1 ± 4.3	0.787 ^a^	24.8 ± 3.9	0.008 ^c^	24.7 ± 4.1	0.088 ^c^	−0.38 ± 0.87	−0.39 ± 1.00	0.787 ^a^
Male waist circumference, cm (n = 75)	89.9 ± 9.4	89.1 ± 11.0	0.720 ^b^	86.7 ± 9.2	<0.001 ^d^	86.2 ± 10.5	<0.001 ^d^	−3.2 ± 3.6	−2.9 ± 3.8	0.683 ^b^
Female waist circumference, cm (n = 13)	85.5 ± 11.7	75.8 ± 6.6	0.239 ^b^	83.7 ± 11.6	0.076 ^d^	76.7 ± 11.0	0.496 ^d^	−1.8 ± 2.3	−1.5 ± 4.6	0.880 ^b^
Systolic blood pressure, mmHg	127.9 ± 19.2	124.3 ± 14.3	0.325 ^b^	125.6 ± 13.1	0.005 ^d^	121.7 ± 11.3	0.122 ^d^	−2.24 ± 14.46	−2.60 ± 12.62	0.896 ^b^
Diastolic blood pressure, mmHg	82.6 ± 13.1	77.9 ± 10.8	0.062 ^b^	77.8 ± 10.7	<0.001 ^d^	74.1 ± 9.0	0.003 ^d^	-4.74 ± 9.69	−3.83 ± 9.26	0.645 ^b^
Triglyceride (TG), mg/dL	138.0 ± 93.3	148.3 ± 199.9	0.234 ^a^	119.2 ± 58.7	0.059 ^c^	128.6 ± 104.3	0.237^c^	−18.87 ± 56.45	−19.67 ± 168.12	0.234 ^a^
High-density lipoprotein (HDL) cholesterol, mg/dL	58.9 ± 13.3	61.5 ± 17.0	0.438 ^a^	63.6 ± 16.9	0.008 ^c^	63.9 ± 16.0	0.032 ^d^	4.68 ± 9.954	2.4 3 ± 8.40	0.438 ^a^
Low-density lipoprotein (LDL) cholesterol, mg/dL	128.7 ± 29.8	124.1 ± 29.1	0.637 ^a^	134.0 ± 30.6	0.097 ^c^	124.1 ± 27.0	0.987 ^d^	5.32 ± 21.82	−0.05 ± 24.92	0.637 ^a^
Glycated hemoglobin A1c (HbA1_c_), %	6.0 ± 0.9	5.7 ± 0.3	0.085 ^a^	5.8 ± 1.3	0.001 ^c^	5.4 ± 0.3	<0.001 ^c^	−0.18 ± 0.64	−0.22 ± 0.21	0.085 ^a^

^1^ Comparing the information and communication technology (ICT) group and control group at baseline; ^2^ comparing the control group at baseline and at 6 months; ^3^ comparing the ICT group at baseline and at 6 months; ^4^ comparing change data between ICT group and control group at 6 months; ^a^ Mann–Whitney U test; ^b^ Student-*t* test; ^c^ Wilcoxon ranked test; ^d^ paired-*t* test.

**Table 2 ijerph-17-08147-t002:** Use of ICT app by ICT group.

Use of ICT Application to Monitor:		<1 Time/Month	1–4 Times/Month	≥5 Times/Month	Mean/Month
Blood pressure (n = 38)	n (%)	8 (21.1%)	7 (18.4%)	23 (60.5%)	7.1
Body weight (n = 47)	n (%)	12 (25.5%)	11 (23.4%)	24 (51.1%)	9.3
Number of steps (n = 47)	n (%)	0	10 (21.3%)	37 (78.7%)	15.1

Information and communication technology (ICT) participants who did not use the app to monitor the relevant indicators were excluded from the table.

**Table 3 ijerph-17-08147-t003:** Comparison of change for ICT participants by frequency of app use.

Measurement		Use of ICT Application to Register *								
		Blood Pressure			Body Weight			Number of Steps		
		Control n = 38	<5 times n = 15	≥5 times n = 23	Control n = 38	<5 times n = 23	≥5 times n = 24	Control n = 38	<5 times n = 10	≥5 times n = 37
Body mass index (BMI), kg/m^2^	Mean	−0.4	−0.1	−0.8	−0.4	−0.2	−0.7	−0.4	−0.1	−0.6
	SD	±0.9	±0.8	±1.1	±0.9	±0.9	±1.1	±0.9	±0.7	±1.1
	*p* value	0.007			0.053			0.093		
Systolic blood pressure, mmHg	Mean	−2.2	−0.1	-6.8	−2.2	−0.9	−5	−2.2	2.4	−5.7
	SD	±14.5	±12.9	±11.3	±14.5	±12.3	±12.9	±14.5	±12.6	±11.8
	*p* value	0.152			0.397			0.126		
Diastolic blood pressure, mmHg	Mean	−4.7	−3.6	−4.2	−4.7	−4.4	−3	−4.7	−3.7	−3.9
	SD	±9.7	±9.3	±9.5	±9.7	±9.6	±8.9	±9.7	±7.8	±10.2
	*p* value	0.79			0.597			0.745		
Waist circumference, cm	Mean	−2.9	−1.5	−4.7	−2.9	−1.6	−4.3	−2.9	−1.7	−3.3
	SD	±3.5	±3.4	±4.2	±3.5	±3.3	±4.4	±3.5	±2.7	±4.5
	*p* value	0.001			0.010			0.161		
Triglycerides, mg/dL	Mean	−0.4	−0.1	−0.8	−0.4	−0.2	−0.7	−0.4	−0.1	−0.6
	SD	±0.9	±0.8	±1.1	±0.9	±0.9	±1.1	±0.9	±0.7	±1.1
	*p* value	0.007			0.053			0.093		
HDL cholesterol, mg/dL	Mean	−2.2	−0.1	−6.8	−2.2	−0.9	−5	−2.2	2.4	−5.7
	SD	±14.5	±12.9	±11.3	±14.5	±12.3	±12.9	±14.5	±12.6	±11.8
	*p* value	0.152			0.397			0.126		
LDL cholesterol, mg/dL	Mean	−4.7	−3.6	−4.2	−4.7	−4.4	−3	−4.7	−3.7	−3.9
	SD	±9.7	±9.3	±9.5	±9.7	±9.6	±8.9	±9.7	±7.8	±10.2
	*p* value	0.79			0.597			0.745		
HbA1c, %	Mean	−2.9	−1.5	−4.7	−2.9	−1.6	−4.3	−2.9	−1.7	−3.3
	SD	±3.5	±3.4	±4.2	±3.5	±3.3	±4.4	±3.5	±2.7	±4.5
	*p* value	0.001			0.010			0.161		

* Person who never used excluded; ICT: information and communication technology; HDL: high-density lipoprotein; LDL: low-density lipoprotein; HbA1c: glycated hemoglobin.

**Table 4 ijerph-17-08147-t004:** Comparing the indicators for the ICT and control groups for whom the program was judged to be effective or where they maintained results.

Indicator	ICT Group	Control Group	*p-*Value *
Body mass index	31 (62.0%)	24 (63.2%)	0.912
Waist circumference	25 (50.0%)	19 (50.0%)	1.000
Systolic blood pressure	38 (76.0%)	14 (36.8%)	0.191
Diastolic blood pressure	44 (88.0%)	29 (76.3%)	0.149
Triglycerides	37 (74.0%)	28 (73.7%)	0.973
HDL cholesterol	47 (94.0%)	37 (97.4%)	0.452
LDL cholesterol	25 (50.0%)	14 (36.8%)	0.218
HbA1_c_	39 (78.0%)	25 (65.8%)	0.203

* Chi-square test; ICT: information and communication technology; HDL: high-density lipoprotein; LDL: low-density lipoprotein; HbA1c: glycated hemoglobin.

**Table 5 ijerph-17-08147-t005:** Comparing insight into health behavior between the ICT and control groups.

Improvement of Lifestyle (n, %)						*p*-Value *
**(1) I am following a diet that is suitable for preventing lifestyle-related diseases.**						
	Strongly agree	Agree	Neither	Disagree	Strongly disagree	0.966
ICT group	4 (9.8%)	24 (58.5%)	8 (19.5%)	3 (7.3%)	2 (4.9%)	
Control group	3 (11.1%)	15 (56.6%)	6 (22.2%)	3 (11.1%)	0	
**(2) I regularly do exercise that is suitable for preventing lifestyle-related diseases.**						
	Strongly agree	Agree	Neither	Disagree	Strongly disagree	0.226
ICT group	5 (12.2%)	27 (65.9%)	4 (9.8%)	1 (2.4%)	4 (9.8%)	
Control group	3 (11.1%)	13 (48.1%)	5 (18.5%)	5 (18.5%)	1 (3.7%)	
**(3) I know my average blood pressure.**						
	Strongly agree	Agree	Neither	Disagree	Strongly disagree	0.197
ICT group	21 (51.2%)	15 (36.6%)	3 (7.3%)	1 (2.4%)	1 (2.4%)	
Control group	8 (29.6%)	17 (63.0%)	1 (3.7%)	0	1 (3.7%)	
**(4)** **I know my average blood glucose level.**						
	Strongly agree	Agree	Neither	Disagree	Strongly disagree	0.980
ICT group	7 (17.1%)	10 (24.4)	6 (14.6)	10 (24.4)	8 (19.5)	
Control group	3 (11.1%)	8 (29.6)	6 (22.2)	5 (18.5)	5 (18.5)	
**(5) I know my average body weight.**						
	Strongly agree	Agree	Neither	Disagree	Strongly disagree	0.188
ICT group	25 (61.0%)	14 (34.1%)	0	1 (2.4%)	1 (2.4%)	
Control group	12 (44.4%)	13 (48.1%)	0	1 (3.7%)	1 (3.7%)	
**(6) I can judge when to start clinical treatment for diabetes, hypertension, dyslipidemia.**						
	Very well	Well	Neither	Not well	Badly/Not at all	0.078
ICT group	4 (9.8%)	12 (29.3%)	12 (29.3%)	3 (7.3%)	10 (24.3%)	
Control group	10 (37.0%)	5 (18.5%)	4 (14.8%)	4 (14.8%)	4 (14.8%)	
**(7) I rate the effectiveness of the program on continuous lifestyle improvement as**						
	Very effective	Effective	Neither	Somewhat ineffective	Not effective at all	0.399
ICT group	1 (2.4%)	21 (51.2%)	8 (19.5%)	2 (4.9%)	9 (22.0%)	
Control group	2 (7.4%)	11 (40.7%)	2 (7.4%)	2 (7.4%)	10 (37.0%)	

* Mann–Whitney U test; ICT: information and communication technology; Diet: based on individual dietary habit, the participant was advised on energy balance (reduce excessive energy intake), nutritional balance, balance fat intake, reduce salt intake, eat 3 times a day, do not eat too late, eat plenty of vegetables and vegetables first, reduce snacks and sugar containing soft drink consumption, and reduce alcohol consumption; Exercise: aerobic exercise of 30–60 min per day, 3 days per week is recommended. However, based on individual lifestyle and medical risk assessment, individually tailored exercise was recommended as to increase the daily activities and walking an extra 10 min.

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
