# Peer review of "A Health Guidance App to Improve Motivation, Adherence to Lifestyle Changes and Indicators of Metabolic Disturbances among Japanese Civil Servants"

_ijerph, 2020, doi:10.3390/ijerph17218147_

Round 1

Reviewer 1 Report

Authors demonstrated the beneficial effects of Apps on promoting the health and metabolic syndrome indices among Japanese adults. Although the effect size is small, the findings are interesting. Authors are advised to address some of the comments given below to strengthen the manuscript.

Comments:

  1. The ‘biological markers in the title is too broad. It would be better if authors could precise the title for the apt meaning of the study.
  2. Abstract, Lines 22-24: Authors stated significant improvement among BMI, male abdominal circumference, SBP, DBP, HDL and HbA1c. Although the results are understandable, HDL should be increased and other outcomes should be decreased. If it is right, please revise the sentence accordingly.
  3. Lines 25-26: Make it as a new sentence, and make sure the usage of ‘however’ and ‘among’.
  4. Introduction: The first 2 references (1 and 2). The references are suitable, however they are very old (2005). There are plenty of relevant studies from that time point. Please update the references with latest reports.
  5. Authors provided very good information about SHG. Such type of assessment or counseling (IHG and MHG) is a very good approach to promote the health of nation. Is it mandatory for every citizen or only for civil servants? In which year this program was initiated in Japan?
  6. Page 2, lines 83-84: Please revise the sentence for clarity.
  7. Authors provided good information about the study participants and ICT program. How often the data were collected from the participants?
  8. It seems authors missed to provide the information about blood sample collection, including time, volume or collection procedure.
  9. How the lipid levels and HbA1c were measured in this study. It’s curious to know whether the participants visited laboratory or participants done their blood check in any designated lab at their respective places?
  10. What is the importance of the Results mentioned in Table 4? Lines 210-211: “The percentages for the control……….” Authors may specify the outcome measures.
  11. Diet: as presented in Table 5, is there any specific diet that is recommended to the participants. If available authors may provide the details.
  12. Similarly, does the exercise completed by the participants was tailored according to the requirements and abilities of the participants?
  13. Please provide the units for BMI, SBP, DBP, LDL, HDL and HbA1c. Also check for other outcome measure and provide the units wherever applicable.
  14. Lines 241-244: Please revise the sentences for the clarity.
  15. Give a space between numbers and words. Please check the whole manuscript for similar corrections and typos.

Author Response

Reviewer 1

Comments and Suggestions for Authors

Authors demonstrated the beneficial effects of Apps on promoting the health and metabolic syndrome indices among Japanese adults. Although the effect size is small, the findings are interesting. Authors are advised to address some of the comments given below to strengthen the manuscript.

Comment 1: The ‘biological markers’ in the title is too broad. It would be better if authors could precise the title for the apt meaning of the study.

Response: Yes, we agree with your observation. We have changed the ‘biological markers’ to ‘indicators of metabolic syndrome’ and precised the title as “A Health Guidance App to Improve Motivation, Adherence to Lifestyle Changes and Indicators of Metabolic Syndrome among Japanese Civil Servants”.

Comment 2: Abstract, Lines 22-24: Authors stated significant improvement among BMI, male abdominal circumference, SBP, DBP, HDL and HbA1c. Although the results are understandable, HDL should be increased and other outcomes should be decreased. If it is right, please revise the sentence accordingly.

Response: Thank you for your useful suggestion. We have revised and rewritten in the ‘Abstract as “After completing the 6-month program, the control group showed a significant decrease in body mass index (p = 0.008), male waist circumference (p < 0.001), systolic blood pressure (BP) (p = 0.005), diastolic BP (p < 0.001), and glycated hemoglobin (HbA1c) (p < 0.001), and increase in high-density lipoprotein (HDL) cholesterol (p = 0.008). However, the ICT group showed a significant decrease in male waist circumference (p < 0.001), diastolic BP (p = 0.003) and HbA1c (p < 0.001), and increase in HDL cholesterol (p = 0.032).”

Comment 3: Lines 25-26: Make it as a new sentence, and make sure the usage of ‘however’ and ‘among’.

Response: We have deleted ‘among’ and made a new sentence as “However, the ICT group showed a significant decrease in male waist circumference (p < 0.001), diastolic BP (p = 0.003) and HbA1c (p < 0.001), and increase in HDL cholesterol (p = 0.032).”

Comment 4: Introduction: The first 2 references (1 and 2). The references are suitable, however they are very old (2005). There are plenty of relevant studies from that time point. Please update the references with latest reports.

Response: We have deleted first two old references and added two recent references in this context.

Comment 5: Authors provided very good information about SHG. Such type of assessment or counseling (IHG and MHG) is a very good approach to promote the health of nation. Is it mandatory for every citizen or only for civil servants? In which year this program was initiated in Japan?

Response: We have added the suggested information accordingly in the main text as “In Japan, all health insurers, whether a national or a social insurance, are required to screen the insured and their dependents aged 40–74 years for risk factors of metabolic syndrome from 2008 [13]” (Page 2, lines 61-62).

Comment 6: Page 2, lines 83-84: Please revise the sentence for clarity.

Response: We have revised the sentence with clarity as “We also examined the changes in indicators of metabolic syndrome related to lifestyle changes made by the participants.”

Comment 7: Authors provided good information about the study participants and ICT program. How often the data were collected from the participants?

Response: We have added the information under the ‘Study protocol’ section (Page 4, lines 159-160).

Comment 8: It seems authors missed to provide the information about blood sample collection, including time, volume or collection procedure.

Response: We agree with your concern. We have now revised and added detail about the blood collection procedure in the ‘Study protocol’ section (Page 4, lines 137-140).

Comment 9: How the lipid levels and HbA1c were measured in this study. It’s curious to know whether the participants visited laboratory or participants done their blood check in any designated lab at their respective places?

Response: We have added the blood check information in the ‘Study protocol’ section (Page 4, lines 137-139).

Comment 10: What is the importance of the Results mentioned in Table 4? Lines 210-211: “The percentages for the control……….” Authors may specify the outcome measures.

Response: Thank you for the suggestion. We have revised and specified the outcome measures in the ‘Results’ section just above the Table 4 (Page 9, lines 212-216).

Comment 11: Diet: as presented in Table 5, is there any specific diet that is recommended to the participants. If available authors may provide the details.

Response: Thank you for the concern of diet. We have revised and added the diet-related information as a note under the Table 5 (Page 11, lines 227-231).

Comment 12: Similarly, does the exercise completed by the participants was tailored according to the requirements and abilities of the participants?

Response: We have also added the exercise-related information in the same note under the Table 5 (Page 11, lines 231-233).

Comment 13: Please provide the units for BMI, SBP, DBP, LDL, HDL and HbA1c. Also check for other outcome measure and provide the units wherever applicable.

Response: Thank you for your valuable advices. We have added the units in Tables and main text wherever applicable.

Comment 14: Lines 241-244: Please revise the sentences for the clarity.

Response: We have revised and rewritten the sentences in the ‘Discussion’ section (Page 11, lines 253-254). 

Comment 15: Give a space between numbers and words. Please check the whole manuscript for similar corrections and typos.

Response: Thank you for your insightful observations. We have checked and corrected all rational space, typos, and necessary formatting.

Thank you for your valuable time, expert review and comments. These are helpful to improve our manuscript.  

Reviewer 2 Report

To Authors:

This study explored whether a health guidance app resulted in sustained motivation, adherence to lifestyle changes and subsequent changes in biological markers among civil servants with metabolic syndrome in Hiroshima Prefecture.

Is the authors ever have tried in their subjects with automatic capturing device instead of entering the values in the app every time?

Overall, this is a very interesting study and useful to make awareness in the community for their health concern. However, the interventions need to be extended in a broader society to make generalize.

As stated in the limitations, the participants were not patients and not randomized, and the sample size is small.

Otherwise, this is a well planned, well executed and well presented study.

Best wishes!

Author Response

Reviewer 2:

Comments and Suggestions for Authors

To Authors:

Comment: This study explored whether a health guidance app resulted in sustained motivation, adherence to lifestyle changes and subsequent changes in biological markers among civil servants with metabolic syndrome in Hiroshima Prefecture.

Response: Thank you for your clarification and understanding.

Comment: Is the authors ever have tried in their subjects with automatic capturing device instead of entering the values in the app every time?

Response: We did not use the automatic capturing device instead of entering the values in the app every time due to limitation of the technical system and budget constraints. We have a plan to try the automatic capturing device in near future. Thank you.

Comment: Overall, this is a very interesting study and useful to make awareness in the community for their health concern. However, the interventions need to be extended in a broader society to make generalize.

Response: Thank you for your appreciation and guidance. We have a plan for further wider interventions, including diverse group of population.   

Comment: As stated in the limitations, the participants were not patients and not randomized, and the sample size is small. Otherwise, this is a well planned, well executed and well presented study.

Best wishes!

Response: Thank you very much for your encouragement and understanding our current limitations.

Reviewer 3 Report

The search for methods that will mobilize people to improve the lifestyle is a very important issue because of the growing problem  of cardiometabolic diseases in developed countries. In my opinion this manuscript is interesting but has some limitations.

The sample size  is relatively small in this study;  for that reason some differences may be statistically insignificant, although an observed decrease or increase may be clinically significant. This problem was included in limitation section. The authors also have listed other limitations that are important and significantly limit the strength of this study.

Beyond the limitations mentioned by the authors, I have found some flaws and enumerate them in the order of appearance.

  1. The topic of this study is focused on metabolic syndrome (MetS). Please specify what criteria for MetS was used?
  2. Line 96 „fasting blood sugar” sugar or glucose? Blood glucose is usually measured.
  3. Line 95 The term „Glucose intolerance” is related to the results of oral glucose tolerance test (OGTT). According to your methodology, the OGTT has not been performed. You measured concentration of blood glucose and the level of HbA1C (reflects the average level of glucose over the past 2 to 3 months). So It would be better to use e.g, elevated glucose concentration
  4. Line 97-98 Please specify the cut-off points for the lipid parameters (HDL, TG, LDL)
  5. Line 99 The inclusion criteria are unclear. Did all the blood parameters have to be outside the normal range to include the individuals to this study?
  6. There is no information about blood pressure in the method section.
  7. Please specify the blood collection procedure. Where the blood measurements were performed (laboratory or POCT methods) at baseline and after follow-up?
  8. Please include the procedures for waist circumference and BP measurements and baseline and after follow-up.
  9. 253 from 355 individuals were excluded from this study. Please specify (in percentage) the reasons of exclusion.
  10. Line 108 -111 The ICT group received the equivalent of the IHG program and the control group, received the equivalent of the usual MHG program. Other programs were used in groups, how this might affect the final results?
  11. Figure 2 Neutral fat . I think it would be better to use triglycerides
  12. Table 1 Male waist circumference n = 30 and Female waist circumference n =5. Is it correctly written?
  13. Table 3 n=15 and n=23 for body weight and number of steps. Is it correctly written?

Author Response

Reviewer 3

Comments and Suggestions for Authors

The search for methods that will mobilize people to improve the lifestyle is a very important issue because of the growing problem of cardiometabolic diseases in developed countries. In my opinion this manuscript is interesting but has some limitations.

The sample size is relatively small in this study;  for that reason some differences may be statistically insignificant, although an observed decrease or increase may be clinically significant. This problem was included in limitation section. The authors also have listed other limitations that are important and significantly limit the strength of this study.

Beyond the limitations mentioned by the authors, I have found some flaws and enumerate them in the order of appearance.

Comment 1: The topic of this study is focused on metabolic syndrome (MetS). Please specify what criteria for MetS was used?

Response: We agree with your concern. We have inserted the explanation and criteria for MetS we used in the ‘Introduction’ part and added ‘Supplemental Table S1’ accordingly, and also added the clarifying information regarding the inclusion criteria.

Comment 2: Line 96 “fasting blood sugar” sugar or glucose? Blood glucose is usually measured.

Response: We have revised this term and rewritten as “fasting blood glucose”.

Comment 3: Line 95 The term “Glucose intolerance” is related to the results of oral glucose tolerance test (OGTT). According to your methodology, the OGTT has not been performed. You measured concentration of blood glucose and the level of HbA1C (reflects the average level of glucose over the past 2 to 3 months). So It would be better to use e.g, elevated glucose concentration

Response: Thank you for your suggestion. We have changed and used as “elevated glucose concentration” in the main text.

Comment 4: Line 97-98 Please specify the cut-off points for the lipid parameters (HDL, TG, LDL)

Response: We mistakenly wrote lipid parameters in criteria. However, it was not listed in the inclusion criteria. We have clearly restated the inclusion criteria with ‘Supplemental Figure S1’.

Therefore, we revised the sentence in the main text under ‘Study participants’ section.

Comment 5: Line 99 The inclusion criteria are unclear. Did all the blood parameters have to be outside the normal range to include the individuals to this study?

Response: We have revised the inclusion criteria to make it clear, and we have added ‘Supplemental Figure S1’ to clarify.

Comment 6: There is no information about blood pressure in the method section.

Response: This study focused on the insurers with risks of metabolic syndrome. Therefore, information about blood pressure was not included. This time, we added the information about blood pressure with the purpose of this prefectural project in the ‘Materials and Methods’ part.

Comment 7: Please specify the blood collection procedure. Where the blood measurements were performed (laboratory or POCT methods) at baseline and after follow-up?

Response: We agree with your concern. We have added the information in ‘Study protocol’ section as “The metabolic indicators were collected by health professionals at baseline and at 6 months (the end of SHG). Before and after SHG, participants visited the same designated licensed health check laboratories, and health professionals did data collection of fasting blood sample (8 ml) and vital signs” (Page 4, lines 137-140).

Comment 8: Please include the procedures for waist circumference and BP measurements and baseline and after follow-up.

Response: We have included the information about the procedures for waist circumference and BP measurements in the “Study protocol’ section as “Waist circumference was measured by placing a tape horizontally around the abdomen at the level of the umbilicus while standing. BP was measured twice at sitting position by the digital BP machine approximately after 5 minutes to calm down, and lower data was used” (Page 4, lines 140-142).

Comment 9: 253 from 355 individuals were excluded from this study. Please specify (in percentage) the reasons of exclusion.

Response: Because of the reason of privacy policy, we were not allowed to see the data of the civil servants (n=6518). Therefore, we obtained the permission for participation from them (individual servants) for viewing their health check data. As a result, only 5.4% gave us permission to see the health check data (even though this was the Hiroshima Prefecture (local government)-led project). Since their insurer, Insurance organization of the local civil servant is a different, independent organization from Hiroshima Prefecture. This is the limitation of National Health Insurance system.

Civil servants of 253 did not match the inclusion criteria. We added the explanation as “Of the 6,518 people in the study population, 355 (5.4%) expressed an interest to participate, and allowed the researchers to obtain their health check results. After the researcher matched their results to the eligibility criteria, only 102 (28.7% of 355) met the criteria (Figure 1).”

Comment 10: Line 108 -111 The ICT group received the equivalent of the IHG program and the control group, received the equivalent of the usual MHG program. Other programs were used in groups, how this might affect the final results?

Response: There were only 14 (4 in ICT group and 10 in control group) individuals who received IHG program out of 88 participants who completed the program. As a result, it is difficult to compare statistically. In addition, all the ICT group received relevant to IHG program, even if they were matched to MHG. For the control group, there was no interventional contact in both IHG (only one midterm interview) and MHG during the program. Therefore, we considered that the IHG and MHG difference do not have to be counted.

Comment 11: Figure 2 Neutral fat. I think it would be better to use triglycerides

Response: We have changed it accordingly as ‘triglycerides’ in Figure 2.

Comment 12: Table 1 Male waist circumference n = 30 and Female waist circumference n =5. Is it correctly written?

Response: Thank you for your critical observations. This time, we have revised and rewritten accordingly in Table 1.

Comment 13: Table 3 n=15 and n=23 for body weight and number of steps. Is it correctly written?

Response: Thank you for your valuable time and checking. We have revised these numbers in Table 3.

Thank you for your valuable time, expert review and comments. These are helpful to improve our manuscript.  

Round 2

Reviewer 3 Report

  1. In my opinion, You should use the definition from 2009 for Asian population e.g. Eating speed and risk of metabolic syndrome among Japanes eworkers: The Furukawa Nutrition and Health Study. Nutrition 78 (2020) 110962    https://doi.org/10.1016/j.nut.2020.110962
  2. BMI, urine glucose, HbA1C are not the indicators of metabolic syndrome (according to the definition). Maybe it will be better to use the term “ indicators of metabolic disturbances” instead of Metabolic Syndrome

Author Response

Comment 1: In my opinion, You should use the definition from 2009 for Asian population e.g. Eating speed and risk of metabolic syndrome among Japanes eworkers: The Furukawa Nutrition and Health Study. Nutrition 78 (2020) 110962. https://doi.org/10.1016/j.nut.2020.110962

Response: Thank you very much for referring this useful publication. With regard to this paper, we have revised the relevant part and adopted in the main text and included the original reference in our manuscript.

Comment 2: BMI, urine glucose, HbA1C are not the indicators of metabolic syndrome (according to the definition). Maybe it will be better to use the term “indicators of metabolic disturbances” instead of Metabolic Syndrome.

Response: Yes, we agree with your observation. We have revised and changed the term “indicators of metabolic disturbances” instead of Metabolic Syndrome in the title and main text wherever applicable. Thank you for your very useful suggestions.

Thank you for your valuable time, expert review and further comments. These are really supporting to improve our manuscript.